# Codelivery of Phytochemicals with Conventional Anticancer Drugs in Form of Nanocarriers

**DOI:** 10.3390/pharmaceutics15030889

**Published:** 2023-03-09

**Authors:** Girish Kumar, Tarun Virmani, Ashwani Sharma, Kamla Pathak

**Affiliations:** 1School of Pharmaceutical Sciences, MVN University, Aurangabad 121105, India; 2Faculty of Pharmacy, Uttar Pradesh University of Medical Sciences, Saifai 206001, India

**Keywords:** cancer, phytochemicals, codelivery, nanotechnology, novel approaches, anticancer drugs

## Abstract

Anticancer drugs in monotherapy are ineffective to treat various kinds of cancer due to the heterogeneous nature of cancer. Moreover, available anticancer drugs possessed various hurdles, such as drug resistance, insensitivity of cancer cells to drugs, adverse effects and patient inconveniences. Hence, plant-based phytochemicals could be a better substitute for conventional chemotherapy for treatment of cancer due to various properties: lesser adverse effects, action via multiple pathways, economical, etc. Various preclinical studies have demonstrated that a combination of phytochemicals with conventional anticancer drugs is more efficacious than phytochemicals individually to treat cancer because plant-derived compounds have lower anticancer efficacy than conventional anticancer drugs. Moreover, phytochemicals suffer from poor aqueous solubility and reduced bioavailability, which must be resolved for efficacious treatment of cancer. Therefore, nanotechnology-based novel carriers are employed for codelivery of phytochemicals and conventional anticancer drugs for better treatment of cancer. These novel carriers include nanoemulsion, nanosuspension, nanostructured lipid carriers, solid lipid nanoparticles, polymeric nanoparticles, polymeric micelles, dendrimers, metallic nanoparticles, carbon nanotubes that provide various benefits of improved solubility, reduced adverse effects, higher efficacy, reduced dose, improved dosing frequency, reduced drug resistance, improved bioavailability and higher patient compliance. This review summarizes various phytochemicals employed in treatment of cancer, combination therapy of phytochemicals with anticancer drugs and various nanotechnology-based carriers to deliver the combination therapy in treatment of cancer.

## 1. Introduction

Despite remarkable progress in treatment of cancer, the prevalence and fatality rates are highest throughout the world [1]. According to a report generated by International Agency for Research on Cancer (IARC) in 2020, approximately 19.3 million new cases and 10 million deaths due to cancer were reported across the world, whilst the new cases and deaths due to cancer in India were 1.32 million and 0.8 million, respectively [2]. This extremely endemic illness is currently thought to be the second most common cause of mortality, having a significant impact on low- and middle-income countries both physically and economically [3]. Minor lifestyle adjustments, reduced alcohol consumption and cessation of tobacco chewing can cut cancer cases by roughly 30 to 50% [4]. Moreover, various treatment strategies such as radiotherapy surgery, immunotherapy, aromatherapy and chemotherapy using drugs are available to cure cancer, but chemotherapy is a very common and widely used treatment option for cancer [5].

The Food and Drug Administration (FDA) has approved more than 300 chemotherapeutic agents to treat cancer, but all these drugs have restricted efficacy in treatment of cancer due to the heterogeneous nature of cancer [6]. Moreover, these drugs also suffer from various hurdles, such as adverse effects, drug resistance, insensitivity of cancer cells to drugs, lack of targeting and patient inconvenience [7,8]. Additionally, chemotherapeutic drugs show their utility in cancer via single targeting of cell proteins, nucleic acids and carcinogenic signalling pathways, which cause adaptation to change the environment by cancerous cells, leading to further development of cancer [6]. Hence, it is required to determine methods to demonstrate their action through multiple targeting of cancer cells so the hurdles associated with conventional chemotherapy can be avoided. Recently, phytochemicals have attracted the attention of researchers as a viable treatment option for cancer due to various attributes: lesser adverse effects, action through multiple targeting, economical, easily available, etc. [9].

Various conventional chemotherapeutic drugs are derived from plants, so researchers are focusing their attention on phytochemicals to treat cancer [10]. These chemotherapeutic agents in monotherapy suffer from various hindrances, which can be combated by employing codelivery of phytochemicals with conventional chemotherapeutics for efficacious treatment of cancer [11,12]. Despite the benefits of combination therapy compared to monotherapy, clinical outcomes are still subpar due to ignorance of poor aqueous solubility, low bioavailability, duration of drugs at the targeting site and the targeting drug [13].

Nanotechnology-based novel carriers, such as nanoemulsion, nanosuspension, nanostructured lipid carriers, solid lipid nanoparticles, polymeric nanoparticles, polymeric micelles or dendrimers, can be employed for codelivery of phytochemicals with conventional chemotherapeutic drugs to elicit optimal clinical outcomes in cancer [14]. These novel carriers have a size range of 1–1000 nm and cause improvement in efficacy of treatment and reduction in adverse effects associated with targeted drugs [6]. These carriers show their potential due to possession of various characteristics of improved solubility, reduced adverse effects, higher efficacy, improved dosing frequency, reduced drug resistance, drug targeting, improved bioavailability and patient compliance [15,16].

## 2. Aetiology, Pathogenesis and Metastasis of Cancer

The terms “carcinogenesis,” “oncogenesis” and “tumorigenesis” refer to the process that causes tumours to form (the “pathogenesis of cancer”), and the agents responsible for causing cancer are carcinogens. Ever since the first carcinogen was discovered, a growing number of substances have been linked to the genesis of cancer. Because of recent enormous advancements in the fields of molecular biology and genetics, there is an increasingly large collection of knowledge regarding the pathogenesis of cancer [17]. According to the World Health Organization (WHO), it is reported that nearly 10 million deaths occurred in 2020 and about one in six deaths were from cancer [18]. Nearly one-third of cancer patients’ deaths occurred by consuming tobacco and alcohol, less consumption of fruits and vegetables in their diet, obesity and lack of exercise. Human papillomavirus (HPV) and hepatitis infections are the major cause of cancer, with approximately 30% of cancer cases reported in lower- and lower-middle-income countries. Breast, lung, colon, rectum and prostate cancers are the most prevalent types of cancer [19]. However, fortunately, many types of cancer are curable with the right diagnosis, early detection and care.

The ability of cancer cells to grow more quickly than healthy cells is one of their defining characteristics. The majority of conventional anticancer drugs are made to target these quickly proliferating cells and stop, kill or slow down their proliferation. Nevertheless, these anticancer drugs also harm or destroy healthy, normal cells. The patient will consequently have serious adverse effects and the effectiveness of the therapy will be limited or diminished [20].

Development of a normal, healthy cell into a tumour cell is known as carcinogenesis, which involves numerous genes and genetic changes. Carcinogenesis is a multiplex process encompassing origination, promotion and progression [21]. The first step involves the beginning of a permanently altered cell and is frequently linked to a mutation and several start pathways. The initial altered cells grow and manifest as a visible mass of cells during the second stage, which is most likely a benign lesion. Epigenetic elements that influence the proliferation of the started cells are undoubtedly present during the promotion stage. It is not well understood exactly how the second stage of carcinogenesis works.

Mostly benign or non-cancerous cells, or occasionally pre-cancerous cells, are the end result of promotion. When these benign cells transition into neoplastic cells, they experience a few additional genetic changes. The third and final stages of carcinogenesis, which involve development of malignant tumours from benign non-cancerous tumours, are distinct from the first two steps [22].

Stem cells play a crucial role in the beginning of carcinogenesis owing to a variety of physical, chemical or biological stimuli, including viruses. The sequential steps are crucial in malignant transformation of preneoplastic cells because such initiated cells would then be exposed to a promotional factor to accelerate full neoplastic cell creation [23]. A multicellular animal’s carcinogenesis process results from numerous cellular chemical, physical, biological or genetic alterations. Although mutation is the primary cause of carcinogenesis, several additional variables also contribute to its growth. Use of either conventional anticancer drugs or plant-derived compounds are available options to reverse or capture the process of carcinogenesis, influencing the carcinogenesis process at each stage of cancer development (Figure 1).

## 3. Progression in Drug Treatment for Cancer

The medical history of cancer dates back thousands of years. The first records of cancer patients come from the ancient Egyptian and Greek cultures, where the disease was mostly treated with radical surgery and cautery, both of which were frequently futile and resulted in patient death [24]. Until the latter half of the 1800s, when the discovery of X-rays and their application to treatment of tumours provided the first modern therapeutic approach in medical oncology, significant discoveries over the centuries enabled identification of the biological and pathological features of tumours. Invention of anticancer drugs and development of chemotherapy for treatment of numerous haematological and solid cancers occurred after the Second World War.

The research for novel drugs to treat cancer has grown exponentially since this epochal turning point. In 1950, the first anticancer drug, antimetabolite, was employed to induce transient remission in children with acute leukaemia, followed by use of an alkylating agent as an anticancer drug after two years. Thereafter, a combination of methotrexate and 6-mercaptopurine was employed for remission of acute leukaemia in children and adults in 1958. In a sequence of the progress of cancer treatment, cyclophosphamide was employed to treat lymphoid leukaemia in 1959. Then, in 1962, vincristine provided promising results in childhood lymphoid leukaemia. After that, a surprise was observed when a birth control drug, tamoxifen, was successfully approved to treat breast cancer in 1978. Thereafter, in 1996, anastrozole obtained approval for treatment of breast cancer. In 1990, the first large molecule, namely rituximab, was introduced as an anticancer drug followed, by introduction of trastuzumab in 1998 to treat breast cancer. Then, imatinib was employed to treat myelogenous leukaemia in 2001, followed by introduction of ipilimumab in 2011 for treatment of metastatic melanoma. In 2014, pembrolizumab was approved as an anticancer moiety. 

## 4. Hindrances in Cancer Treatment by Conventional Drugs

Various anticancer drugs are available to treat cancer, but these drugs pose restrictions in therapeutic efficacy for treatment of cancer due to various reasons, such as non-specific treatment, higher adverse effects, required higher doses, small half-life, wide biological distribution, drug resistance, drug instability and lack of drug targeting [25]. These drugs destroy non-cancerous cells along with cancerous cells at a high pace, leading to severe adverse effects, which are a major cause of the increased death rate among cancer patients.

Progression of resistance by cancerous cells to anticancer drugs is a main hindrance in efficient treatment of cancer, which occurs due to various mechanisms, such as drug efflux, epigenetics, DNA mutation, cell death inhibition, drug degradation, drug inactivation and drug target alteration [26,27]. The requirement of higher doses of anticancer drugs leads to increased toxicities in non-cancerous cells and multiple-drug resistance [28]. Furthermore, drug instability and low aqueous solubility of anticancer drugs also cause obstacles in treatment of cancer [29]. Drugs approved by the FDA to treat cancer have been summarized in Table 1, along with their adverse effects. Various hindrances in treatment of cancer employing conventional anticancer drugs in monotherapy have depicted in Figure 2.

## 5. Herbal Treatment of Cancer (Phytochemical-Based Treatment of Cancer)

Because of the limitations of conventional chemotherapeutic drugs, there is an urgent need for novel cancer treatment. Recently, plant-derived agents, namely phytochemicals, have drawn the attention of researchers as a new treatment modality for cancer owing to various attributes of less adverse effects, action through multiple pathways and cost-effectiveness [2,10]. From ancient times, humans have made great use of plants for treatment of various kinds of ailments, including cancer [41]. Since most available conventional chemotherapeutic drugs, such as vincristine, vinblastine and paclitaxel, are plant-derived, the attention of researchers turned towards phytochemicals for treatment of cancer [42].

According to studies, there are over 250,000 plant species in the plant kingdom, but only about 10% of those have demonstrated their potential as a treatment option for various kinds of diseases, depicting that a vast portion of plant species are yet to be explored, which could cause a revolution in treatment of cancer [9]. Various phytochemicals and their derivatives are present in diverse parts of plants, such as seeds, flowers, bark, fruit, leaf, embryo and rhizomes [43,44]. In addition, these phytochemicals and their derivatives possess various pharmacological properties, such as anti-inflammatory, antimicrobial, antifungal, antihypertension, antiaging, antioxidant, immunomodulator, antimalarial, anticancer, etc.

Various plant-derived products, such as flavonoids, alkaloids, terpenes, vitamins, glycosides, minerals, oils, gums, biomolecules and other metabolites (primary or secondary), have proven their anticancer potential owing to various mechanisms: inhibition of cancer-cell-activating proteins, enzymes such as cyclooxygenase, topoisomerase, CDK2, Cdc2, CDK4 kinase, MMP, MAPK/ERK, signalling pathways, activation of DNA repair mechanism, induction of antioxidant action or stimulation of formation of protective enzymes (Caspase-3, 7, 8, 9, 10, 12) [45,46] (Figure 3).

Various preclinical studies have demonstrated that a combination of phytochemicals with conventional anticancer drugs is more efficacious than phytochemicals individually to treat cancer because plant-derived compounds have lower anticancer efficacy than conventional anticancer drugs.

Various phytochemicals with the potential to improve anticancer activity in codelivery include curcumin, resveratrol, genistein, epigallocatechin gallate, allicin, quercetin, thymoquinone, piperine, naringenin, naringin, emodin, luteolin, β-carotene, anthocyanins, berberine, ursolic acid, withaferin A, sulforaphane and colchicine [13,47,48] (Figure 4).

Although phytochemicals have huge potential as anticancer drugs, they also suffer from various limitations, such as low solubility, poor bioavailability, high dose, narrow therapeutic index, fast absorption by normal cells, high apparent volume of distribution leading to accumulation of drugs in normal cells, high clearance rate and short elimination half-life [13,49]. Phytochemicals also have the potential to improve anticancer properties of other chemotherapeutic agents by decreasing their adverse effects [50,51]. Hence, these days, plant-derived drugs or phytochemicals are used in combination with conventional chemotherapeutic agents for efficacious treatment of cancer with low adverse effects [52]. To date, no combination of conventional anticancer drug with phytochemicals is available as an anticancer treatment, but curcumin in monotherapy is in clinical trials on the market as an anticancer drug. In NCT01294072, a phase I randomized clinical trial was conducted to study the ability of plant exosomes to deliver curcumin to normal or colon cancer tissues, enrolling 35 participants. The status of clinical trial was recruiting. The primary outcome measure of the study was to compare concentrations of curcumin in normal tissues and cancerous tissues after 7 days of ingestion; the secondary outcome measures were safety and tolerability of curcumin alone as determined by adverse events after 7 days of enrolment [53].

## 6. Nanocarrier-Based Codelivery of Chemotherapeutic Agent with Phytochemicals

To address the obstacles associated with administration of conventional chemotherapeutic agents and phytochemicals in monotherapy, codelivery of these agents has emerged as an imperative approach, resulting in enhanced therapeutic efficacy in cancer and reduced adverse effects [54]. Codelivery of drugs in cancer is advantageous due to possession of various attributes, including reduced number of doses leading to patient compliance, reduction in multiple-drug resistance and decreased drug doses, leading to reduction in adverse effects in non-cancerous cells [55]. Moreover, various research findings have illustrated that codelivery of chemotherapeutic drugs with phytochemicals is advantageous in terms of synergistic anticancer effects, reversing multiple-drug resistance and reduction in adverse effects [6] (Figure 5). Phytochemicals diminish augmentation and metastasis of cancerous cells along with increasing sensitivity of cancerous cells to apoptosis and DNA destruction caused by chemotherapeutic agents [56,57].

Codelivery of phytochemicals with chemotherapeutic agents provides a reduction in chemoresistance developed by the reduction in drug uptake by cancerous cells, stimulation of DNA repair mechanism, uncontrolled expression of drug-resistant proteins and overexpression of carriers responsible for higher outflow of drug [6,58]. Moreover, chemotherapeutic agents in monotherapy are required in larger doses to elicit anticancer activity, which leads to severe adverse effects [50], such as cardiotoxicity, nephrotoxicity, ototoxicity and hepatotoxicity [34,59,60]. Additionally, codelivery of antioxidants with chemotherapeutic drugs may result in notable toxicity reductions so that more patients can complete prescribed chemotherapy regimens, improving the likelihood of success in terms of tumour response and survival [61]. Hence, codelivery of phytochemicals with chemotherapeutic agents is not only responsible for anticancer activity and reversal of chemoresistance but also reduces adverse effects linked with chemotherapeutic agents. Ni W et.al, prepared curcumin with 5-fluorouracil-loaded nanoparticles to provide synergistic effects in hepatocellular carcinoma [62] and Elkashty, O.A. and Tran, S.D investigated the synergistic effect of sulforaphane with 5-fluorouracil in which dose of 5-fluorouracil was reduced with improved cytotoxic effects, leading to reduced adverse effects [63].

Despite the benefits of codelivery of phytochemicals with chemotherapeutic agents, the results are insignificant [64] due to various reasons of low aqueous solubility, poor bioavailability, lack of drug targeting to a cancerous cell and duration of targeting at cancerous cell [65], and drug targeting to a particular cancerous cell with reduced adverse effects is the main challenge faced during codelivery of phytochemicals with chemotherapeutic agents. It is mainly due to the presence of highly organized physical, physiological and enzymatic barriers, which results in limited drug partitioning and distribution to the target site and nonselective tissue toxicity in combination therapies [66,67].

Furthermore, codelivery of phytochemicals with chemotherapeutic drugs is suboptimal due to various physiochemical and pharmacodynamic characteristics of different drug molecules, lack of optimistic dosing and scheduling of various drugs in codelivery, hydrophobicity of the drug, first-pass effect, low aqueous solubility and poor bioavailability [6]. Moreover, codelivery of small drug molecules shows more adverse effects clinically. In addition, the differences in pharmacological fate and pharmacokinetic profile of individual agents may cause serious side effects and systemic toxicity. These hindrances associated with codelivery of chemotherapeutic drugs with phytochemicals prompted development of novel drug carriers, which mainly include nanotechnology-based drug carriers termed nanocarriers [68].

Nanocarriers are a potential option for codelivery of drugs in treatment of cancer due to various attributes of drug targeting at the desired site, biodegradability, increased dosing interval, reduction in adverse effects, reduction in dose, nanosize, improved stability and inability to deliver hydrophilic as well as hydrophobic drugs [69]. Owing to their nanosize, nanocarriers can cross various physiological hurdles and accumulate the drug in sufficient amounts by the targeted cancerous cell, which leads to improvement in bioavailability of the drugs and avoidance of adverse effects in healthy cells [66,70]. These are more efficient to deliver two or more drugs together. A drug with anticipated pharmacokinetic and pharmacodynamic characteristics can be administered using nanocarriers employing modification in size and shape of the nanocarriers [71]. These enable the improvement in therapeutic efficacy of drugs with reduction in adverse effects [72]. Codelivery in nanocarriers enclosed the pharmacokinetics of the drugs, which enables unifying of pharmacokinetic properties of the codelivered drugs, increased biodistribution time and enhanced selectivity to the tumour. The remarkable advantage of nanocarriers is the ability to release therapeutic agents in a controlled manner in terms of location, time, amount and sequence. Codelivery systems can be considered potential candidates to maximize treatment efficiency, minimize side effects and improve the pharmacokinetic profile of combined therapeutic agents [73]. Furthermore, they provide controlled, sustained and targeted release of the embedded drugs. The half-life of encapsulated drugs can be increased in blood circulation.

Functionalization of nanocarriers employing stimuli responsiveness, such as pH, temperature, time and decoration of nanocarrier’s surface with specific ligands, can be provided, which elicits prolonged drug retention at targeted site as well as improved cellular uptake of targeted drugs. Functionalization of nanocarriers leads to increase in bioavailability of targeted drugs. The ligands employed for functionalization include antibodies, aptamers, small molecules, peptides, etc. [74]. Codelivery of conventional anticancer drug paclitaxel with naringin employing polymeric micelles improved in vitro cytotoxicity against MCF-7 breast cancer cells and enhanced internalization of paclitaxel. In this, naringin serves as chemosensitizer, improving the lethal effect of paclitaxel in prostate cancer synergistically [75]. Codelivery of doxorubicin with curcumin improved anticancer potential of doxorubicin along with reduction in adverse effects. To date, numerous conventional anticancer drugs are codelivered with plant-derived compounds to improve their efficacy [76].

Various nanocarriers, solid lipid nanoparticles, nanostructured lipid carriers, nanoemulsions, polymeric nanoparticles, polymeric micelles, liposomes, dendrimers, carbon nanotubes, metallic nanoparticles and nanoemulsions have been utilized for codelivery of anticancer drugs owing to their ability to entrap the drugs followed by release on targeted site [77,78] (Figure 6). Moreover, these also protect drug molecules from hazardous environmental factors, which can cause gastrointestinal degradation of the drugs [79]. Modification in shape, size and surface properties of nanocarriers can be performed to elicit maximum efficiency, which leads to improved drug efficiency, decreased adverse effects, avoidance of multiple-drug resistance and maximization of drugs in targeted cells [80]. Various nanocarriers for codelivery of conventional anticancer drugs with phytochemicals have been discussed in preceding section.

### 6.1. Solid Lipid Nanocarriers (SLNs)

Researchers have focused much attention on lipid nanoparticles because they are at the forefront of the fast-evolving field of nanotechnology and hold great promise for achieving the objective of controlled and targeted drug delivery in cancer treatment [81]. SLNs provide various noteworthy benefits of improved solubility, low adverse effects, improved bioavailability of drugs, adaptability of encapsulation of both hydrophilic and hydrophobic drugs, improved stability, specificity and probability of large-scale production [82,83].

The properties of biodegradability and biocompatibility of SLNs make them less toxic than other nanocarriers, such as polymeric nanoparticles [84]. Nanosize (less than 400 nm), easy functionalization, chemical and mechanical stability and increased delivery of lipophilic phytochemicals are more advantageous characteristics of SLNs [85]. SLNs also enable to overcome several physiological barriers that hinder drug delivery to cancerous cells and are also able to escape multidrug resistance mechanisms characteristic of cancerous cells. SLNs have the distinct inherent capacity to concentrate the drug in cancerous cells precisely due to their properties of increase in permeability and retention time [86]. When SLNs are phagocytized at the cancerous site, the drug is delivered closer to the intracellular site of action, leading to an increase in cell internalization [54]. SLNs deliver drugs to the targeted cancerous site due to various mechanisms, such as active mechanisms and passive mechanisms.

These are composed of solid lipids or a mixture of lipids and surfactants. Moreover, aqueous phase, surface modifiers, cosurfactants stealthing agents and cryoprotective agents may also be present in their structure [87]. The hydrophobic drug or combination of hydrophobic drugs is entrapped in the solid lipid matrix of SLNs, enabling protection of drugs from chemical degradation, which leads to physical stability. These improve the half-life of drugs in blood circulation and modify their release pattern, which leads to an increase in therapeutic efficiency of anticancer drugs [88]. Wang L. et al. prepared paclitaxel- and narigenin-loaded SLNs to treat glioblastoma multiforme in which SLNs were functionalized using cyclic RGD peptide sequence to improve drug targeting to cancerous site. It was found that pharmacokinetic parameters, such as C_max_, T_max_ and relative bioavailability, of peptide functionalized SLNs were improved compared to plain SLNs as well as drug suspension. Moreover, functionalized SLNs possessed improved cytotoxicity compared to free drug suspension on U87MG glioma cells [89].

Pi C. et al. fabricated SLNs of curcumin and paclitaxel to treat lung cancer. It was observed that SLNs of combination provided improved area under the curve (AUC), prolongation of drug residence time and increase in half-life of the drugs, resulting in long circulation time in systemic circulation. Furthermore, the rate of lung tumour suppression was 78.42% using SLNs of combination of paclitaxel and curcumin, whilst it was 40.53% and 51.56% using paclitaxel and combination (paclitaxel and curcumin), respectively [90]. Despite various advantages of SLNs in cancer treatment, these also possessed some limitations of poor drug loading capacity, expulsion of drug, increased incidence of polymorphic transitions and unpredictable agglomeration, which must be addressed [91].

### 6.2. Nanostructured Lipid Carriers (NLCs)

To address various above-mentioned limitations of SLNs, NLCs have proven their efficacy as advanced drug carriers in cancer treatment. Their broad relevance as drug carriers is due to their distinctive characteristics, which include increased drug encapsulations, long-term chemical and physical stability of the encapsulated drug, surface modifications and site-specific targeting [92]. These possess liquid lipids along with solid lipids in their structure, which provides imperfections in the lipid matrix. These imperfections cause prevention of drug leakage during prolonged storage, resulting in improved drug loading [93]. The presence of liquid lipids along with solid lipids in NLCs enables accumulation of a large number of drugs compared to solid lipids and liquid lipids individually [16]. Drug bioavailability can be improved by NLCs, which results in improved drug transport through the intestine and protection of drugs from the hazardous environment of the gastrointestinal tract [94].

Moreover, NLCs enable drug targeting through the lymphatic system, resulting in various advantages, such as avoidance of first-pass metabolism, decreased hepatotoxicity and improved bioavailability [95]. Alhalmi A. et al. codelivered raloxifene and naringin, employing NLCs for treatment of breast cancer. It was found that NLCs of dual drugs provided 2.1 and 2.3 times improved permeability profiles of naringin and raloxifene than their suspension. Furthermore, it was observed that codelivery of raloxifene with naringin in NLCs reduced the acute toxicity of raloxifene, which could be attributed to the antioxidant property of naringin [16]. Zhao X. et al. delivered doxorubicin with curcumin in form of NLCs to treat liver cancer, and improved cytotoxicity and reduced inhibitory concentration were observed in HepG2 and LO2 cells. Furthermore, Annexin-V-fluorescein isothiocyanate/propidium iodide double staining demonstrated increased apoptosis in HepG2 cells treated with doxorubicin and curcumin-loaded NLCs compared to free doxorubicin and doxorubicin nanoparticles [76].

### 6.3. Liposomes

Due to possession of various characteristics, such as the capacity to encapsulate high doses, possibility to deliver hydrophilic and hydrophobic drugs, increase in circulation time of drug, biodegradability, biocompatibility, improved durability, low adverse effects, controlled drug delivery, increased rate of dissolution, the capability of drug targeting to individual cells, easy manufacturing and versatility, liposomes have emerged as a potential carrier for codelivery of anticancer drugs [96,97]. These are spherical-shaped vesicles composed of phospholipids and cholesterol bilayers, resulting in creation of two microenvironments, which enable codelivery of the drugs [98]. These have a size range of 0.025 to 2.5 µm [99]. The amount of encapsulation of drugs in liposomes is governed by size and number of bilayers along with size of vesicles [100]. Liposomal structures can be modified to elicit desired therapeutic effects [101].

Liposomal entrapped drugs can be targeted to a desired site by active and passive mechanisms. Passive targeting of liposomes enables accumulation of drugs preferentially in cancerous cells through enhanced permeability and retention property (EPR). Active targeting of liposomes to the desired site can be provided using functionalization of liposomal surface to various kinds of antibodies, which leads to an increase in specificity to cancerous site. Aside from the capability to target drugs by active and passive mechanisms, liposomes also can facilitate release of drugs in specific tumour cells under influence of pH, light, sound and enzymes [102]. In addition, liposomal efficiency at the cancerous site can be improved using external stimuli, such as temperature, pH and ultrasound, triggering release of drugs in the interstitium after concentrating in the desired site [90].

Otherwise, functionalization of the liposomal surface with PEG causes improved efficiency of anticancer drugs at the targeted site owing to an increase in drug circulation time [103]. Moreover, liposomes are less taken by GIT, heart and tissues, which leads to a decrease in adverse effects [104]. Cheng Y. et al. codelivered cisplatin and curcumin in form of nanoliposomes for efficient treatment of hepatocellular carcinoma. Codelivery of drugs in form of nanoliposomes exhibited improved anticancer property against HepG2 tumour cells, with IC50 value of 0.62 micro M. It also provided improved ROS levels intracellularly during treatment of HCC cells. Furthermore, it provided prolonged retention time of 2.38 h compared to individual drug formulations and improved anticancer effect in animal hepatoma H22 and human xenograft model along with reduced adverse effects [105].

### 6.4. Polymeric Nanoparticles

Possession of various important features of biocompatibility, biodegradability, smaller size, increased surface volume ratio and easier modification of structure and surface, polymeric nanoparticles (PNPs) have been extensively used for codelivery of anticancer drugs with phytochemicals. Moreover, PNPs protect entrapped drug molecules and controlled or sustained the release of entrapped drugs [106]. The potential of PNPs to deliver anticancer drugs is continuously increasing due to the inability to target the drugs only on cancerous cells.

PNPs are composed of natural, semisynthetic and synthetic polymers, which are either biodegradable or non-biodegradable. The main characteristic of PNPs for drug targeting in cancer is their size, which must be below 100 nm due to the inability to pass through apertures in the endothelial of cancerous cells [107]. Second, the shape of PNPs also plays a vital role in the efficient delivery of anticancer drugs to the target site. The shape of PNPs must be spherical because spherical drug particles are effectively taken by the targeted cancerous cells [108].

To enhance the circulation time of the drug and minimize the drug interactions with blood proteins, coating with polyethene glycol can be employed. PEGylation of PNPs causes an increase in the half-life of the drugs in the blood and leads to improvement in the stability of drug molecules. Further, PEGylation increases the hydrophilicity of drug molecules and enables the PNPs to encapsulate hydrophilic and lipophilic drugs, which release the drugs in a controlled manner [109]. Interestingly, PNPs can be functionalized with various molecules, such as folic acid, and antibodies to elicit more selectivity for cancerous cells.

Various polymers used for preparation of PNPs include natural (gelatin, lysozyme, cellulose, chitosan, dextran, albumin, collagen), semisynthetic (methylcellulose) and synthetic polymers (polylactic acid (PLA), poly lactide-co-glycolide (PLGA), thiolated poly methacrylic acid) [110]. PLGA is a biodegradable polymer which is approved by the FDA for drug targeting in the treatment of cancer. The acceptability of PLGA-based nanoparticles is mainly due to their hydrolysis in the body, during which it metabolizes in monomer units glycolic acid and lactic acid, which ensures their reduced toxicity [111]. PNPs can provide the controlled and targeted release of drugs at cancerous sites owing to response to various stimuli (pH, temperature), which trigger the release of drugs at the desired site [112].

Amjadi S. et al. delivered doxorubicin and betanin via encapsulation in PEGylated gelatin nanoparticles. These PNPs were made pH-responsive using methoxy polyethene glycol-poly 2-dimethylamino ethyl methacrylate-co-itaconic acid to trigger the release of the drug in a controlled way at the desired site. It was found that PNPs of doxorubicin and betanin reduced the cell sustainability amount of MCF-7 cells in breast cancer more than doxorubicin and betanin alone [113]. Hu H. et al. delivered paclitaxel and curcumin using PLGA nanoparticles and was found that optimized formulation provided improved cytotoxicity, having reduced IC_50_ in MCF-7 cells of breast cancer compared to free drugs [114].

### 6.5. Dendrimers

A dendrimer is a nanometric, multibranched, star-shaped polymeric vesicle that looks like a tree. It consists of branches interiorly, a central core and various functional groups exteriorly [115]. The presence of various branches on the surface of the dendrimer enables codelivery of various drugs [116]. Due to the possession of a low polydispersity index, controlled molecular weight and improved biocompatibility, dendrimers have emerged as drug carriers in cancer treatment. The functional groups present on the exterior surface of dendrimers enable the entrapment of a combination of drugs in dendrimers. These functional groups can be modified to provide drug targeting at the specific cancerous site. Moreover, drug delivery using dendrimers causes improved aqueous solubility, stability, bioavailability of drugs, reduced adverse effects, loading of higher dose, enhanced drug efficacy and drug release in controlled as well as sustained manner.

Drug entrapment in dendrimers is possible due to mechanisms of physical interaction and chemical interaction [117]. In physical interaction, the drug is entrapped into dendrimers by non-covalent bonds, whilst in chemical interaction drug is covalently attached to dendrimers [118,119]. Various anticancer drugs, such as methotrexate, cisplatin, 5-fluorouracil, paclitaxel and doxoroubicin, have been delivered successfully employing dendrimers along with reduced adverse effects [118].

Various dendrimers employed for codelivery of anticancer drugs with phytochemicals include polyamidoamine (PAMAM), poly-L-lysine (PPL) and polypropylene imine (PPI) amongst PAMAM dendrimers have been extensively utilized for drug delivery of anticancer drugs due to hydrophilic nature, biocompatibility and non-immunogenicity [120]. Despite showing various benefits as drug carriers, dendrimers show hemolytic and cytotoxic properties, which raises a major question about the safety of dendrimers [121]. These toxic effects can be reduced using surface functionalization of functional groups present on the exterior surface of dendrimers. Surface functionalization of dendrimers can be performed using polyethene glycols, which increases drug circulation time owing to EPR besides reduction in toxic effects [118,122].

Ghaffari M. et al. developed PAMAM dendrimers for codelivery of curcumin with Bcl-2 siRNA against HeLa cells of cancer. These dendrimers provided improved cellular uptake and greater inhibition of cancer cell proliferation than PAMAM curcumin nanoformulation and plain curcumin drugs [123].

### 6.6. Polymeric Micelles (PMs)

PMs have appeared as versatile drug carriers in the era of nanocarriers due to possession of various characteristics of increased aqueous solubility of the drug, marvellous biocompatibility, enhanced permeability and reduced toxic effects [124,125]. In addition, PMs cannot modify the drug release and concentrate it on targeted cancerous sites [126]. Due to their size in the nanometric range, they are prone to accumulate in the microenvironment of cancer through EPR [127,128].

Numerous combinations of anticancer drugs have been delivered employing PMs to improve the synergistic effect of combined drugs but unfortunately, the traditional PMs provided limited synergistic effect due to non-selectivity and incomplete release behaviour of the drugs. These limitations have prompted the development of modified PMs, which provide drug targeting to the cancerous site using active and passive mechanisms, triggering the microenvironment of cancer using specific stimuli, such as light, pH, ultrasound and temperature [129].

PMs can be fabricated using amphiphilic di or triblock copolymers. The hydrophilic portion of copolymer includes polymers such as PEG and poly N-isopropylacrylamide, whilst the hydrophobic portion includes polypropylene glycol (PPG), poly caprolactone (PCL) [130].

Sabra S.A. et al. codelivered rapamycin and wogonin in form of polymeric micelles prepared by hydrophilic lactoferrin and hydrophobic zein. Codelivery of drugs provides increased circulation time and targeting to specific cancer cells. Moreover, crosslinking by glutaraldehyde was observed, which provided improved stability and reduced size. PMs provided a fast release of wogonin, which enabled the inhibition of efflux pump resulting in potentiation of targeting of rapamycin to cancerous site [131].

### 6.7. Nanoemulsions (NEs)

Researchers have shifted their attention towards nanoemulsions due to unique properties such as physical stability, higher surface area, prolonged circulation time, amphiphilicity, specific drug targeting, tumour imaging properties, optical clarity, biodegradability, improved aqueous solubility and bioavailability. Moreover, nanoemulsions can be surface modified to enable passive and active targeting of the drugs [132,133].

NEs are colloidal dispersions of two immiscible liquids stabilized by amphiphilic surfactants. These are in the nonmetric size range of 20–200 nm [134,135]. Due to nanosize in addition to possession of active and passive mechanisms, NEs are enable to accumulate in the cancer microenvironment and overcome various associated obstacles [136]. NEs functionalization is possible using conjugation with various antibodies for targeting precise sites. It has been evaluated that the conjugation of anticancer drugs with antibodies results in the incorporation of drugs in cancerous cells for the successful delivery of the drugs to the targeted site [137]. Furthermore, conjugation of drug with antibody can be made responsive to stimuli to cause more specificity towards cancerous cells.

Various anticancer drugs have been codelivered employing NEs to improve the therapeutic efficacy and bioavailability of the drugs. Ganta S and Amiji M delivered combination of paclitaxel and curcumin in form of nanoemulsion to SKOV3 cancer bearing mice. The paclitaxel exhibited 4.1-fold improved AUC when administered in nanoemulsion form to curcumin treated mice. Relative bioavailability of paclitaxel was 5.2-fold greater, which resulted in 3.2-fold improved accumulation of paclitaxel in cancer tissues [138].

### 6.8. Carbon Nanotubes (CNTs)

Owing to various characteristics such as reduced size, increased surface area, high drug loading capability, controlled and sustained release of the drugs and drug targeting have focused the considerable attention of researchers towards CNTs as a potential drug carrier to deliver anticancer drugs. Moreover, the presence of numerous sites at the surface of CNTs facilitates the delivery of more than one drug at a time [139].

CNTs are mainly of two types namely single-walled carbon nanotubes (SWCNTs) and multi-walled carbon nanotubes (MWCNTs) amongst MWCNTs are more prominent recently as drug carriers [140]. MWCNTs possess considerable absorptive surface for anticancer drugs, which can be targeted to the specific cancerous site [141]. CNTs should be functionalized employing different polymers, chemical groups or biomolecules to ensure their targeting capacity and safety in cancer treatment owing to improvement in hydrophilicity and reduction in cytotoxicity properties of CNTs [142]. Functionalization of CNTs surface can be carried by covalent and non-covalent bonding of various types of polymers and chemical groups at the surface of CNTs [143].

PEG, the most popular FDA-approved polymer has been extensively used for the surface functionalization of CNTs to impart increased solubility and biocompatibility. Monoclonal antibodies also can be conjugated with CNTs for efficient treatment of cancer [144]. Arginylglycylaspartic acid (RGD) can also be employed for surface functionalization of CNTs resulting in active drug targeting to the cancerous site [145]. In addition, recently carbohydrate-based polymers, such as lactose and mannose, also have been employed for surface functionalization of CNTs to provide drug targeting to desired cancerous sites [146].

Raza K. et al. fabricated MWCNTs of docetaxel and piperine with a view of increased tissue permeation, bioavailability and anticancer activity and was found that MWCNTs of the conjugate of both drugs provided 6.4 times improved AUC than pure drugs [147].

### 6.9. Metallic Nanoparticles (MNPs)

Because of their rich surface functionalization, lengthy activity period, relatively narrow size and shape distribution and the ability for optical or heat-based treatment techniques, MNPs are particularly alluring in nanomedicine for targeting therapeutic agents in cancer. Owing to their higher density, MNPs can be easily absorbed by cells, which is helpful for cancer control strategies [148]. MNPs have also been claimed to enable superior targeting, gene silencing and drug delivery, particularly when functionalized with targeting ligands that allow regulated deposition into cancerous cells [149].

MNPs can alter the microenvironment of a tumour by transforming unfavourable circumstances into ones that can be used therapeutically. For instance, external stimuli such as light, heat, ultrasonic waves and magnetic fields might improve the capacity of MNPs to target biological systems by changing their redox potential and producing reactive oxygen species (ROS) that further sensitise target tissues [150].

Various MNPs employed to treat numerous kinds of cancers include gold nanoparticles (Au NPs), silver nanoparticles (Ag NPs), iron oxide nanoparticles (IONPs) and zinc oxide nanoparticles (Zn ONPs) [151]. Au NPs possess several desirable characteristics, including low toxicity, immunogenicity, great stability, improved biocompatibility, increased permeability, increased retention and easily functionalized surface [152]. Other extensively studied nanoparticles are Ag NPs, which are alluring in cancer treatment due to possession of various attributes, such as unique physicochemical and biological characteristics, including biocompatibility, high surface-to-volume ratio, powerful antibacterial activity, outstanding surface plasmon resonance, ease of functionalization and cytotoxicity against cancer cells [148,153]. Ag NPs can modify autophagy of cancer cells whether they work as cytotoxic agents by themselves, in combination with transported compounds or in conjunction with other therapies [154].

IONPs have attracted specific attention in emerging magnetic nanoparticles due to possession of excellent targeting abilities under an external magnetic field [155]. In particular, IONP-based delivery systems that are injected move via blood capillaries to the appropriate spot when an external magnetic field is applied, releasing the medicine in cancerous cells and boosting therapeutic efficacy without harming nearby normal cells [156].

Hiremath C. et al. developed oleic-acid-coated IONPs stabilized by folic-acid-modified pluronicF127 for codelivery of curcumin and paclitaxel in breast cancer. It was found that cytotoxic property of folic-acid-modified NPs was greater and was further improved on application of external magnetic field [157]. Various nanoformulations for codelivery of conventional anticancer drugs and phytochemicals have been summarized in Table 2.

## 7. Conclusions

Cancer is a multistep and multifactorial disease whose prevalence and mortality rate are increasing with time. That is why treatment of cancer became a major challenge. Recently, surgery is mainly used as a treatment for cancer in association with chemotherapy. However, chemotherapy has low proficiency as a treatment due to various obstacles, such as adverse effects, drug resistance, insensitivity of cancer cells to drugs, lack of targeting and patient inconvenience. Codelivery of anticancer drugs with phytochemicals can provide better therapeutic efficiency owing to synergism in treatment of cancer than anticancer drugs in monotherapy. Recently, codelivery of conventional anticancer drugs with phytochemicals has paid attention to treatment of cancer, but occurrence of overlapping, multiple and occasionally unanticipated adverse effects is a great challenge in codelivery of anticancer drugs. On this subject, encapsulation of drugs provides safer combination. Further, incorporation of nanotechnology-based delivery systems, such as solid lipid nanoparticles, nanostructured lipid carriers, liposomes, polymeric nanoparticles, nanoemulsions, dendrimers, polymeric micelles, metallic nanoparticles, or carbon nanotubes, can elicit maximal therapeutic efficiency to alleviate cancer due to possession of improved solubility, reduced adverse effects, higher efficacy, improved dosing frequency, reduced drug resistance, drug targeting, improved bioavailability and patient compliance. Deep knowledge or understanding of mechanisms for cancer development as well as nanoformulations to deliver anticancer drugs along with phytochemicals offer a new option for treatment of cancer. Codelivery of anticancer drugs with phytochemicals has demonstrated outstanding performance in experiments, but clinical efficiency is lacking owing to deficiency in optimization of precise combination of drugs, such as sequence of drug exposure and ratio of drugs. Hence, more thorough and effective pharmacodynamic evaluation techniques are required to justify the rationality of codelivery of anticancer drugs with phytochemicals in nanoformulations.

## Figures and Tables

**Figure 1 pharmaceutics-15-00889-f001:**
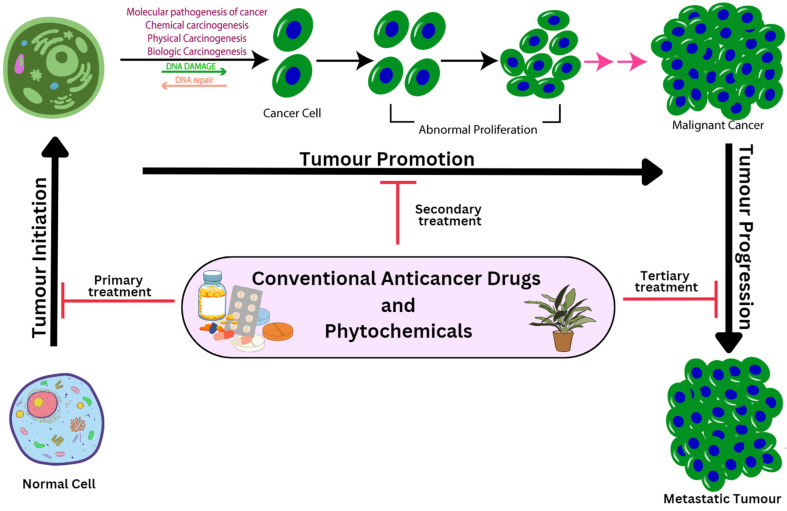
Various stages for progression of cancer along with their treatment approaches.

**Figure 2 pharmaceutics-15-00889-f002:**
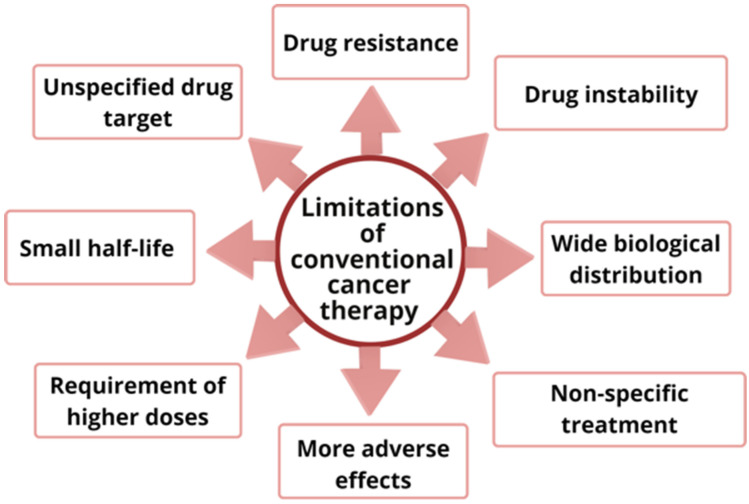
Various hindrances associated with conventional anticancer drugs.

**Figure 3 pharmaceutics-15-00889-f003:**
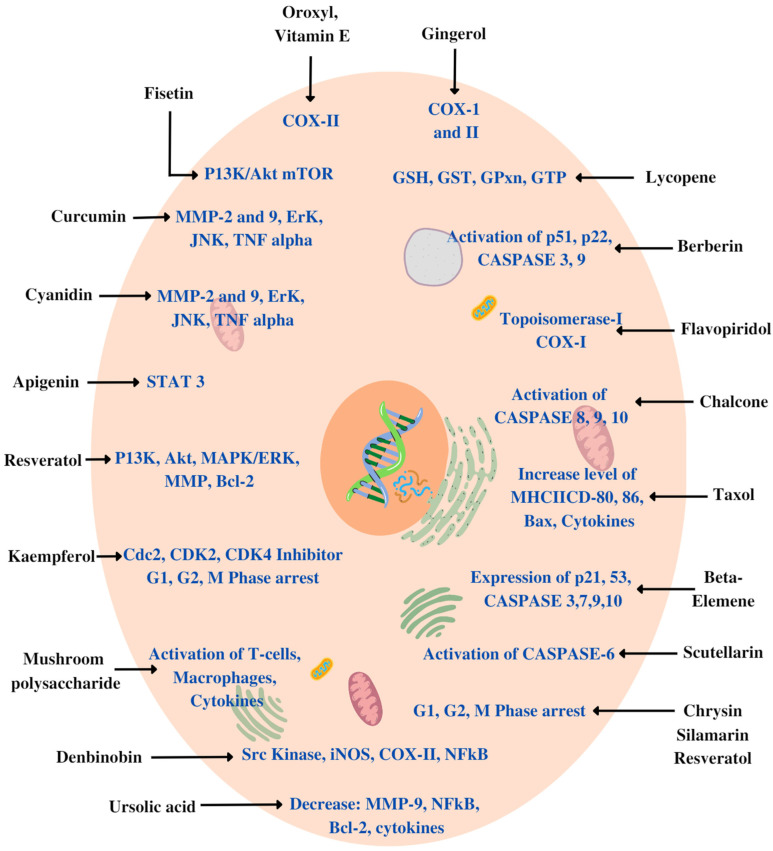
Various mechanisms for working of phytochemicals as an anticancer agent.

**Figure 4 pharmaceutics-15-00889-f004:**
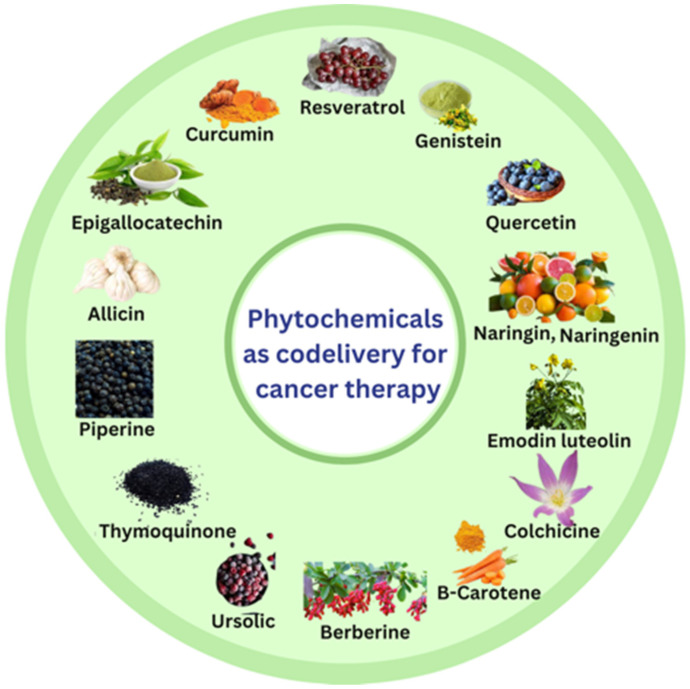
Various phytochemicals employed as codelivery with anticancer drugs.

**Figure 5 pharmaceutics-15-00889-f005:**
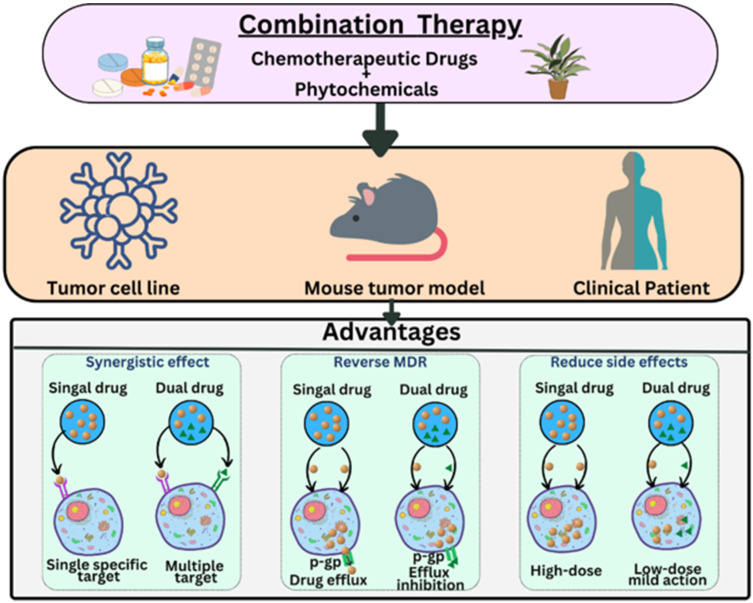
Advantages of codelivery of anticancer drugs with phytochemicals.

**Figure 6 pharmaceutics-15-00889-f006:**
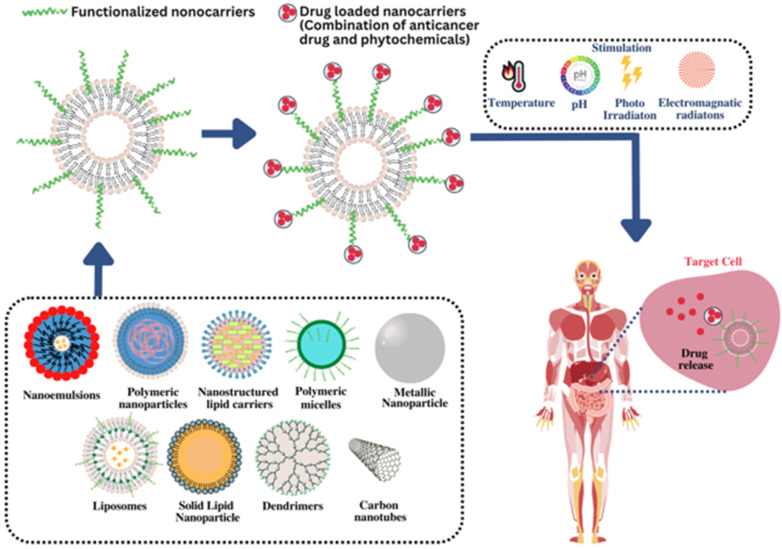
Mechanism of various nanocarriers to target the drug at the cancer site.

**Table 1 pharmaceutics-15-00889-t001:** List of anticancer drugs along with their adverse effects.

Drug Category	Sub-Class	Examples	Adverse Effects	References
Alkylating agent	Nitrogen mustard	Cyclophosphamide	Myelosuppression, alopecia hemorrhagic cystitis and gonadal damage	[30]
	Platinum agents	Cisplatin, carboplatin, oxaplatin	Nephrotoxicity, neurotoxicity, ototoxicity, myelosuppression	[31]
Antimetabolites	Pyrimidine derivatives	5-Fluorouracil, floxuridine, gemcitabine, capecitabine	Leukopenia, pulmonary embolism, neutropenia, pyrexia, thrombocytopenia and diarrhoea	[32]
	Antifolates	Methotrexate	Hepatotoxicity, nausea, vomiting, loss of appetite, mucosal disorders	[33]
Antimitotic agents	Taxanes	Docetaxel, paclitaxel, cabazitaxel	Febrile neutropenia, infusion reactions, fluid retention and fatigue	[34]
	Vinca alkaloids	Vincristine, vinblastine	Neurotoxicity, chest pain, acute cardiac ischemia, acute pulmonary effects, hand-foot syndrome, hepatic and pulmonary toxicity	[35]
Topoisomerase inhibitors	Topoisomerase-I inhibitors	Topotecan, irinotecan	Myelosuppression GI toxicity, cardiotoxicity, alopecia, secondary leukaemia	[36]
	Topoisomerase-II inhibitors	Etoposide		
Antitumor antibiotics	Anthracyclines	Doxorubicin, daunorubicin, epirubicin, idarubicin, valrubicin	Cardiotoxicity, alopecia, tissue necrosis	[37,38]
	Others	Bleomycin, mitoxantrone	Alopecia, bone marrow suppression, febrile neutropenia, cardiotoxicity	[39]
Hormonal drugs	Selective estrogen receptor modifier	Tamoxifen, Raloxifen	Hot flashes, vaginal dryness, depression, weight gain, sleep disturbances	[40]

**Table 2 pharmaceutics-15-00889-t002:** List of nanoformulations for codelivery of anticancer drugs with phytochemicals.

Type of Nanoformulation	Anticancer Drug	Phytochemical	Type of Cancer	Result Outcomes	References
SLNs	Paclitaxel	Naringenin	Glioblastoma multiforme	Exhibited 1.7–2.8 times improved C_max_ and AUC_0−t_ for both paclitaxel and naringenin from the SLNs than their drug suspension whilst no significant change in T_max_ was observed. Moreover, cyclic RGD-modified SLNs possessed more improved drug absorption than plain SLNs. It also exhibited improved cytotoxicity than drug suspension.	[89]
SLNs	Paclitaxel	Curcumin	Lung cancer	Exhibited 1.40 and 2.28 times improved AUC for curcumin and paclitaxel, respectively, provided 6.94 and 6.46 times extended residence time for curcumin and paclitaxel, respectively, achieving long circulation. The rate of tumour suppression of SLNs was 78.42% higher than 40.53% and 51.56% for paclitaxel and a combination of curcumin with paclitaxel.	[90]
SLNs	5-fluorouracil	Curcumin	Liver cancer	SLNs of curcumin with layered double hydroxide 5-fluorouracil provided a synergetic effect on SMMC-7721 cells more strongly than plain drugs in combination. FACS analysis exposed that SLNs of combination prompted 80.1% apoptosis in SMMC-7721 cells.	[158]
SLNs	Docetaxel	Curcumin	Breast cancer	Possessed a noteworthy improvement in AUC of 594.21 ± 64.34 μg/mL h than 39.05 ± 7.41 μg/mL h of Taxotere^®^ and MRT of 31.14 ± 19.94 h than 7.24 ± 4.51 h of Taxotere^®^. Moreover, the accumulation of docetaxel was reduced in the heart and kidney compared to Taxotere^®^. Targeting efficiency towards MCF-7 cells was also revealed using fluorescence microscopy.	[159]
NLCs	Tamoxifen citrate	Coenzyme Q10	Breast cancer	Revealed increased % cell viability for normal WISH cell line reaching 100% at 0.25 μg/mL. The lipid nanocarrier exhibited LC 50 on the MCF-7 cell line of 1.6 μg/mL as compared to 4.8 μg/ml on the WISH cell line.	[54]
NLCs	Paclitaxel	α-Tocopherol succinate	Retinoblastoma	Possessed anticipated physicochemical properties and might lead to an efficacious therapeutic option to treat retinoblastoma.	[160]
NLCs	Tamoxifen	Sulforaphane	Breast cancer	Provided 5.2- and 4.8-fold improved oral bioavailability of tamoxifen and sulforaphane along with reduction in tamoxifen-associated toxicity in vivo.	[161]
NLCs	Doxorubicin	Lapachone	Breast cancer	Exhibited improved retention of doxorubicin on MCF-7 ADR cells. In vivo studies on MCF-7 ADR tumour-bearing animal models exhibited improved efficacy.	[162]
NLCs	Docetaxel	Curcumin	Lung cancer	Demonstrated significantly improved cytotoxic activity towards NCI-H460 cells.	[163]
NLCs	Temozolomide	Curcumin	Brain cancer	Demonstrated accumulation of drugs at brain and cancer sites. The inhibitory effect is due to arresting of the S phase cell cycle along with induced apoptosis. Moreover, the toxic effects were absent at normal doses.	[164]
NLCs	Doxorubicin	Β-element	Lung cancer	Displayed improved cytotoxicity, synergistic antitumor effect and insightful tumour inhibition ability.	[165]
NLCs	Docetaxel	Curcumin	Lung cancer	Provided considerably improved apoptotic, anti-proliferative, anti-angiogenic and anti-metastatic activities than Taxotere^®^. NLCs displayed considerably reduced adverse effects of docetaxel	[166]
Lipid chitosan hybrid nanoparticles	Cisplatin	Curcumin	Ovarian cancer	Provided significant cytotoxicity than cisplatin-loaded nanoparticles as well as curcumin-loaded nanoparticles after 48 hours of treatment.	[167]
Hybrid nanoparticles	Cisplatin	Oleanolic acid	Gastric carcinoma	Exhibited Induced tumour cells apoptosis, reduced adverse effects and reversal of multidrug resistance	[168]
Hybrid nanoparticles	Methotrexate	Beta-carotene	Breast cancer	Provided the highest apoptosis index against MCF-7 cells. Moreover, methotrexate-induced renal and hepatic toxicity was reduced by codelivery of beta-carotene.	[169]
Hybrid nanoparticles	Docetaxel	Resveratrol	Lung cancer	Exhibited significant synergistic potential along with best cancer inhibition ability and minimal systemic toxicity.	[170]
Hybrid nanoparticles	Cisplatin	Curcumin	Cervical cancer	Exhibited significantly improved cytotoxic effects and demonstrated the highest anticancer potential compared to other formulations.	[171]
Hybrid nanoparticles	Doxorubicin	Gallic acid	Leukemia	Displayed protruding cytotoxicity and the best synergistic effect. Nanoparticles revealed improved inhibition of tumour growth.	[172]
Calcium carbonate nanoparticles	Cisplatin	Oleanolic acid	Hepatocellular carcinoma	Exhibited improved HepG2 cell apoptosis along with alleviation of drug-induced hepatotoxicity.	[173]
Nanosponge particles	Tamoxifen	Quercetin		Provided alleviation of the hepatotoxicity produced during the treatment along with improvement in the uptake of tamoxifen.	[174]
Lipid nanoparticles	Doxorubicin	Curcumin	Hepatocellular carcinoma	Demonstrated synergistic activity on the apoptosis, proliferation and angiogenesis of hepatocellular carcinoma. Moreover, the mRNA levels of *MDR1*, *bcl-2* and *HIF-1α* and protein levels of P-gp, Bcl-2 and HIF-1α were reduced.	[175]
Ph-sensitive galactosylated nanoparticles	Sorafenib	Curcumin	Hepatocellular carcinoma	Possessed the smallest tumour volume of 239 ± 14 mm^3^ along with an inhibition rate of 77.4% employing pH-sensitive lactosylated nanoparticles.	[176]
Dual targeting nanoparticles	5-fluorouracil	Curcumin	Hepatocarcinoma	Exhibited synergistic antitumor efficiency established by cytotoxicity and animal studies. These provided improved cellular uptake and stronger cytotoxicity for cancer cells.	[62]
PLGA nanoparticles	Methotrexate	Curcumin	Breast cancer	Exhibited 2.5 and 1.7 fold lower IC50 values after 24 and 48 h, respectively, than methotrexate nanoparticles. The cytotoxic property was greater than in other formulations and tumour incidence and size were reduced in the case of PLGA nanoparticles entrapped with both methotrexate and curcumin than other formulations.	[177]
PLGA nanoparticles	Salinomycin	Curcumin	Breast cancer	Provided higher efficacy of CD44 cell surface glycoprotein-functionalized PLGA nanoparticles against breast cancer stem cells by the convincing arrest of G_1_ cell cycle and restraining epithelial–mesenchymal transition.	[178]
PLGA nanoparticles	Topotecan	Thymoquinone		Demonstrated the sustained release of both the drugs, having a minimal burst release and a total percentage release of more than 90% in 96 h.	[179]
PLGA nanoparticles	Tamoxifen	Quercetin	Breast cancer	Exhibited improved efficiency revealed by increased cellular uptake, nuclear co-localization and cytotoxicity in MCF-7 cells. Provided 5- and 3-fold improved oral bioavailability for tamoxifen and quercetin, respectively. Possessed a higher rate of tumour suppression against a DMBA-induced breast cancer model.	[180]
PLGA nanoparticles	Doxorubicin	Resveratrol	Breast cancer	Exhibited noteworthy cytotoxicity on MDA-MB-231/ADR cells and MCF-7/ADR cells. Moreover, co-encapsulated nanoparticles delivered the drugs to cancer tissue.	[181]
PLGA-PEG nanoparticles	Gemcitabine	Betulinic acid	Solid tumor	Provided increased cytotoxicity than native drugs solution. Moreover, suppression of tumour growth was more efficient in the solid tumour model than the native gemcitabine and betulinic acid at the same concentrations.	[182]
PLA nanoparticles	Sorafenib	Plantamajoside	Hepatocellular carcinoma	Provided improved anticancer effect of sorafenib on hepatocellular carcinoma cells due to reversal of drug resistance.	[183]
PLA nanoparticles	Daunorubicin	Glycyrrhizic acid	Leukemia	Exhibited a tremendous synergistic effect leading to ominously greater cell inhibition. Cell apoptosis was improved but did not influence MDR1 expression.	[184]
PLGA nanoparticles	Doxorubicin	Berberine	Breast cancer	Exhibited significant improvement in mitochondrial membrane permeability along with the arrest of progression of the cell cycle at the sub-G1 phase.	[185]
PLGA nanoparticles	Paclitaxel	Curcumin	Brain cancer	Provided synergistic effect on inhibition of cancer growth via cell cycle arrest and apoptosis induction. Efficient brain deposition of the drug was demonstrated.	[186]
PEGylated nanoparticles	Paclitaxel	Dihydroartemisinin	Colorectal cancer	Exhibited improved apoptosis in colorectal HT-29 cells. Moreover, nanoparticles displayed significantly improved accumulation in the cancer site due to the increased permeability and retention effect.	[187]
Polymeric nanoparticles	Doxorubicin	Curcumin	Lymphoma	Exhibited improved intracellular delivery along with increased cytotoxic effect, induced sophisticated rates of apoptosis in BJAB cells. BJAB cells provided inhibited tumour growth than doxorubicin alone.	[188]
Polymeric nanoparticles	Paclitaxel	Curcumin	Breast cancer	Provided substantial inhibition of cancer growth with elongated survival time along with reduced adverse effects.	[189]
Polymeric nanoparticles	Paclitaxel	Resveratrol		Demonstrated improved anticancer effect along with improvement in the sensitivity of multidrug-resistant cancer cells to the drug.	[190]
PLGA-PEG-PLGA polymeric nanoparticles	5-Fluorouracil	Chrysin	Colon cancer	Exhibited considerably improved growth inhibitory activities in the HT29 cell line.	[191]
Polymeric nanoparticles	Doxorubicin	Curcumin		Exhibited improved cytotoxicity against HCT-116 cells. Cellular uptake of drugs was improved via active targeting.	[192]
Polymeric nanoparticles	Doxorubicin	Epigallocatechin gallate	Gastric cancer	Exhibited internalization into gastric tumour cells via CD44 ligand recognition and subsequent inhibition of cell proliferation.	[193]
Polymeric nanoparticles	Paclitaxel	Silybin	Breast cancer	Demonstrated effective accumulation of the drug in tumour site and inhibition of tumour growth as well as sensitization effect of silybin on paclitaxel cytotoxic chemotherapy.	[194]
Self-assembled Ph-sensitive nanoparticles	Methotrexate	Ganoderma lucidum Polysaccharide	Breast cancer	Demonstrated improved cancer suppressive activities with fewer adverse effects.	[195]
Gold nanoparticles	Methotrexate	Curcumin		Provided increased cytotoxic effect against C6 glioma and MCF-7 cancer cell lines along with high hemocompatibility. It also possessed active targeting proficiency against MCF-7 cancer cells due to the presence of the “antifolate” drug methotrexate.	[196]
Gold nanoparticles	Doxorubicin	Resveratrol	Cervical cancer	Provided strong deposition of the drug in the tumour cells.	[197]
Gold nanoparticles	Doxorubicin	Epigallocatechin-3-gallate	Prostate cancer	Exhibited inhibition of the proliferation of PC-3 tumour cells along with the enzyme-responsive intracellular release of doxorubicin.	[198]
Hybrid nanoparticles	Paclitaxel	Curcumin		Provided significantly improved early and late apoptosis along with induction of a stronger G_2_/M arrest and significantly increased subG_1_ cell population	[199]
Hybrid nanoparticles	Doxorubicin	Curcumin		Exhibited improved cytotoxicity against A549 cells along with increased cellular uptake.	[200]
Mixed polymeric micelles	Paclitaxel	Naringin	Breast cancer	Exhibited increased intracellular uptake along with 65% in vitro cytotoxicity against breast cancer cells at its lower dose of 15 µg/Ml	[75]
Micelles	Doxorubicin	Curcumin		Exhibited strongest cytotoxic properties as well as improved cell apoptosis-inducing activities against doxorubicin-resistant MCF-7/Adr cells.	[201]
Polymeric micelles	Docetaxel	Resveratrol	Breast cancer	Provided a stronger synergistic effect, elongated release profiles and improved cytotoxicity in MCF-7 cells.	[202]
Polymeric micelles	Triptolide	SN-38	Gastric cancer	Provided reduced Cancer-associated fibroblasts activity and inhibited Cancer-associated fibroblasts, induced proliferation, migration and chemotherapy resistance of gastric cells.	[203]
Polymeric micelles	Doxorubicin	Curcumin		Exhibited induced apoptosis.	[204]
Polymeric micelles	Paclitaxel	Capsaicin	Breast cancer	Provided prolonged circulation time and privileged tumour tissue buildup compared to the taxol solution. Micelles displayed greater antitumor activity.	[205]
PCM micelles	Paclitaxel	Cucurbitacin B	Gastric cancer	Exhibited reduced tumour but the loss of bodyweight was not significant.	[206]
Nanomicelles	Dooxorubicin	Rhein	Ovarian cancer	Demonstrated improved cytotoxicity and increased apoptosis-inducing actions in SKOV3/DOX cells. Micelles displayed better targeting ability towards cancer along with reduced toxicity.	[207]
Nanomicelles	Docetaxel	Curcumin	Ovarian cancer	Demonstrated stronger inhibition and proapoptotic activities on A2780 cells. Micelles provided inhibition of tumour proliferation, suppression of tumour angiogenesis and promotion of tumour apoptosis.	[208]
Nanomicelles	Gemcitabine	Camptothecin	Breast cancer	Provided superior accumulation of nanomicelles at the cancer site, which could enhance therapeutic activity and reduce side effects.	[209]
Self-assembling micelles	Dooxorubicin	Honokiol	Glioma	Displayed inhibition of glioma growth more ominously. Micelles increased the antitumor effect of doxorubicin by increasing tumour cell apoptosis, suppressing tumour cell proliferation and inhibiting angiogenesis.	[210]
Responsive micellar system	Paclitaxel	Curcumin	Breast cancer	Demonstrated the maximum level along with achieving greater tumour inhibition effect.	[211]
Nano liposomes	Cisplatin	Curcumin	Hepatocellular carcinoma	Exhibited improved anticancer activity against HepG2 cells having the IC_50_ of 0.62 Μm. Moreover, provided increased intracellular ROS levels during the HCC cell treatment. It also demonstrated the prolonged retention time and increased antitumor effect.	[105]
Liposomes	Doxorubicin	Curcumin		Provided distinct inhibition of tumour growth in mice. Inhibition of tumour growth was 2–3 fold less in mice than in formulations having drugs individually.	[212]
Liposomes	Irinotecan	Berberine	Pancreatic cancer	Exhibited significant inhibition of tumour growth in the BXPC-3 pancreatic cancer model than Onivyde and decreased the gastrointestinal toxicity in mice caused by irinotecan.	[213]
Liposomes	Doxorubicin	Berberine	Triple-negative breast cancer	Provided significant inhibition of tumour growth in 4T1 murine mammary carcinoma model than Doxil and completely combat the myocardial rupture toxicity caused by Doxil in mice.	[214]
Liposomes	Doxorubicin	Pachymic acid and dehydrotumulosic acid	Breast cancer	Exhibited significantly increased anticancer effect of doxorubicin in cancer-bearing mice than other monotherapy groups.	[215]
Liposomes	Cisplatin	Curcumin	Breast cancer	Exhibited 10 times greater apoptosis than liposomes of cisplatin only. Codelivery of cisplatin liposomes with curcumin decreased the viability of breast cancer cells by 82.5%.	[216]
Liposomes	Doxorubicin	Schisandrin B	Lung cancer	Demonstrated improved cytotoxicity, improved cardiotoxicity and inhibition of the invasion and metastasis of tumours.	[217]
Liposomes	Combretastatin A4 phosphate	Curcumin	Liver cancer	Exhibited improved cytotoxicity and increased accumulation in the tumour site. Moreover, liposomes displayed stronger inhibition of tumour proliferation.	[218]
Nanoemulsion	Paclitaxel	Curcumin	Ovarian cancer	Demonstrated effective delivery of drug intracellularly in both SKOV3 and SKOV3(TR) cells. Administration of curcumin inhibits NFkappaB activity and down-regulates P-glycoprotein expression in resistant cells.	[138]
Double nanoemulsion	5-fluorouracil	Curcumin	Breast cancer	Demonstrated improved cytotoxicity against the MCF-7cells.	[219]
Multiwalled CNTs	Docetaxel	Piperine	Breast cancer	Demonstrated improved anticancer activity and stronger cytotoxicity in MCF-7 cells	[147]
Multiwalled CNTs	N-Desmethyl tamoxifen	Quercetin	Gastric cancer	Exhibited decreased IC_50_ values along with improved cellular uptake in drug-resistant MDA-MB-231 cells. The drug availability in blood circulation was also increased.	[220]

## Data Availability

Not applicable.

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
