# Peer review of "Codelivery of Phytochemicals with Conventional Anticancer Drugs in Form of Nanocarriers"

_pharmaceutics, 2023, doi:10.3390/pharmaceutics15030889_

Round 1
Reviewer 1 Report
The authors introduced the DDS to co-deliver phytochemicals and conventional anticancer drugs. It is suitable for pharmaceutics. But please address the following points.
1. Please discuss the advantage of co-delivery nanosystem, such as unifying pharmacokinetics of different drugs.
2. For small molecule combination therapy, always more side effects will be observed in clinic. This is another reason to use co-delivery nanosystem.
3. Any example of nanosystem in clinic is co-delivery of phytochemicals and conventional anticancer drugs?
4. Please give some specific figures as typical co-delivery examples. By the way, please include another phytochemical, colchicine. There is some reports on co-delivery (10.1002/adma.202105254).
5. The authors should discuss the common advantages of DDS, such as functionality (10.1021/jacs.0c09029).
6. It is better to discuss more deeply on how phytochemical can affect the co-delivery system, particularly using some specific anticancer drugs, such as cisplatin (10.1016/j.jconrel.2022.03.049). The authors can discuss their own opinions in concluding remarks.
Author Response
Reviewer 1
- Please discuss the advantage of co-delivery nanosystem, such as unifying pharmacokinetics of different drugs.
Ans: Thank you for insightful and valuable suggestion of reviewer. We have incorporated the above point in manuscript as follows:
The differences in pharmacological fate and pharmacokinetic profile of individual agents may cause serious side effects and systemic toxicity. The codelivery in nanocarriers enclosed the pharmacokinetics of the drugs which enables the unifying of pharmacokinetic properties of the codelivered drugs, increased biodistribution time, and enhanced selectivity to the tumor. The remarkable advantage of nanocarriers is the ability to release therapeutic agents in a controlled manner in terms of location, time, amount, and sequence. Therefore, co-delivery systems can be considered potential candidates to maximize treatment efficiency, minimize side effects, and improve the pharmacokinetic profile of combined therapeutic agents. Furthermore, they provide controlled, sustained, and targeted release of the embedded drugs. The half life of the encapsulated drugs can be increased in the blood circulation.
- For small molecule combination therapy, always more side effects will be observed in clinic. This is another reason to use co-delivery nanosystem.
Ans: Thanks for providing insightful and valuable suggestions. Your point is absolutely correct as we have perceive the idea about the same by keeping in mind your above mentioned statement also. The codelivery of phytochemicals with chemotherapeutic drugs is suboptimal due to various physiochemical and pharmacodynamic characteristics of different drug molecules, lack of optimistic dosing and scheduling of various drugs in codelivery, hydrophobicity of the drug, first-pass effect, low aqueous solubility, poor bioavailability. Moreover, codelivery of small drug molecules shows more adverse effects clinically. These hindrances associated with the codelivery of chemotherapeutic drugs with phytochemicals prompted the development of novel drug carriers which mainly include nanotechnology-based drug carriers termed nanocarriers.
We have incorporated above point in the manuscript under Section 6.
- Any example of nanosystem in clinic is co-delivery of phytochemicals and conventional anticancer drugs?
Ans. We have also worked on the same at the time of writing the manuscript. But, after going through extensive literature review we have not found any nanosystem for codelivery of phytochemicals with conventional anticancer drugs clinically. However various nanosystem for codelivery of conventional anticancer drugs are available clinically which is given in below mentioned table for your reference purpose.
|
Name |
Composition |
Status |
Indication |
|
CPX-1 |
Irinotecan and floxuridine |
Phase 2 |
Colorectal cancer |
|
DPX-0907 |
Liposomal 7 tumor-specific |
Phase 1 |
Ovarian, breast, and prostate |
|
MM-302 |
HER2-targeted liposomal |
Withdrawn |
Breast cancer |
|
Dher2 + AS15 |
Truncated HER2 protein |
Phase 2 |
Breast cancer |
- Please give some specific figures as typical co-delivery examples. By the way, please include another phytochemical, colchicine. There is some reports on co-delivery (10.1002/adma.202105254).
Ans: Thank you for valuable suggestions. We have gone through suggested article which is very impactful and knowledgeable. We have incorporated colchicine in phytochemicals for codelivery in Section 5 of manuscript and Figure 4.
- The authors should discuss the common advantages of DDS, such as functionality (10.1021/jacs.0c09029).
Ans: Thank you for your insightful suggestion. This will definitely improve our manuscript. As per your suggestion, we have discussed about the functionality of the nanocarriers as given in suggested article as follows:
Functionalization of nanocarriers employing stimuli responsiveness like pH, temperature, time and decoration of nanocarriers surface with specific legends can be provided which elicit prolonged drug retention at targeted site as well as improved cellular uptake of targeted drugs. The functionalization of nanocarriers leads to increase in bioavailability of the targeted drugs. The ligands employed for functionalization includes antibodies, aptamers, small molecules, peptides etc
- It is better to discuss more deeply on how phytochemical can affect the co-delivery system, particularly using some specific anticancer drugs, such as cisplatin (10.1016/j.jconrel.2022.03.049). The authors can discuss their own opinions in concluding remarks.
Ans: As per your suggestions we have incorporated text regarding some specific anticancer drugs such as paclitaxel, cisplatin, doxorubicin as following:
In a study, codelivery of conventional anticancer drug paclitaxel with naringin employing polymeric micelles improved in vitro cytotoxicity against MCF-7 breast cancer cells and enhanced internalization of paclitaxel. In this, naringin serves as chemosensitizer, improving the lethal effect of paclitaxel in prostate cancer synergistically. In another study, codelivery of doxorubicin with curcumin improved anticancer potential of doxorubicin along with reduction in adverse effects. Till date, numerous conventional anticancer drugs are codelivered with plant derived compounds to improve their efficacy.
As per your suggestions, following text has been incorporated in conclusion.
Recently the codelivery of conventional anticancer drugs with phytochemicals has paid attention to the treatment of cancer but the occurrence of overlapping, multiple, and occasionally unanticipated adverse effects is a great challenge in codelivery of anticancer drugs. On this subject, encapsulation of drugs in nanocarriers provides the safer combination

Reviewer 2 Report
The authors generally wrote an excellent paper on phytochemicals and anti-cancer drug delivery. It would be nice to clarify the configuration and publish it after major revision.
1.In the title " in form of nanocarriers" Are they both carriers?
2.It is an undeniable fact that plant-derived compounds have low anti-cancer efficacy (compared with anticancer drugs). It's not just issues like solubility (To be drugs). It is better to write the contents in the text and abstract.
3. It would be better to drastically cut Chapter 2 (Aetiology, pathogenesis, and metastasis of cancer) in half. It is difficult to see that it is directly related to the text.
4. Chapter 3 needs to be rewritten in relation to "phytochemicals". Can phytochemicals solve the drawbacks described in Chapter 4?
5. Figure 1 also needs to be modified (related to "phytochemicals" and "nanocarrier?"). Information on the occurrence of simple cancers is not required.
6. Chapter 5 is good, but it is needed to break up the paragraphs and clarify the content for the potential readers.
7. Check the grammar of the words in Figure 2 again.
8. Overall, I think it will help to understand if I divide the paragraphs and describe them.
9. It would be good to reflect the classification (nanocarriers) in Chapter 6 in Table 2 as well. The contents of Table 2 are good, but the classification is awkward. For example, What does simple nanoparticle (Table 2) mean (not SLN?, NLC?) ?
Author Response
Reviewer 2
1.In the title " in form of nanocarriers" Are they both carriers?
Ans: Both (anticancer drugs and phytochemicals) are the drug moieties which are encapsulated in nanocarriers in combination to provide improved anticancer efficacy.
2.It is an undeniable fact that plant-derived compounds have low anti-cancer efficacy (compared with anticancer drugs). It's not just issues like solubility (To be drugs). It is better to write the contents in the text and abstract.
Ans: Yes, Your above mentioned statement is absolutely correct. Plant derived compounds have low anti-cancer efficacy when compared to conventional anticancer drugs. We have incorporated the following point in text as well as in the abstract as per your valuable suggestion. Your suggestion will definitely improve our manuscript.
Various preclinical studies have demonstrated that the combination of phytochemicals with conventional anticancer drugs is more efficacious than phytochemicals individually to treat cancer due to the fact that plant derived compounds have low anticancer efficacy than conventional anticancer drugs. Moreover, phytochemicals suffer from poor aqueous solubility and reduced bioavailability which must be resolved for efficacious treatment of cancer
- It would be better to drastically cut Chapter 2 (Aetiology, pathogenesis, and metastasis of cancer) in half. It is difficult to see that it is directly related to the text.
Ans: Thank you for valuable and insightful suggestions of reviewer. We have cut the chapter 2 and tried to relate it with the text.
The terms "carcinogenesis," "oncogenesis," and "tumorigenesis" refer to the process that causes tumours to form (the "pathogenesis of cancer") and the agents responsible for causing cancer are carcinogens. Ever since the first carcinogen was discovered, a growing number of substances have been linked to the genesis of cancer. Because of recent enormous advancements in the fields of molecular biology and genetics, there has been an even bigger collection of knowledge regarding the pathogenesis of cancer[17]. According to WHO, it is reported that nearly 10 million deaths occurred in 2020 and about one in six deaths[18]. Nearly one-third of cancer patients' deaths occurred by consuming tobacco, and alcohol, less consumption of fruits and vegetables in their diet, obesity, and lack of exercise. Human papillomavirus (HPV) and hepatitis infections are the major cause of cancer, with approximately 30% of cancer cases reported in lower and lower-middle countries. Breast, lung, colon, rectum, and prostate cancers are the most prevalent types of cancer[19]. But fortunately, many types of cancer are curable with the right diagnosis, early detection and care.
The ability of cancer cells to grow more quickly than healthy cells is one of their defining characteristics. The majority of conventional anticancer drugs are made to target these quickly proliferating cells and stop, kill, or slow down their proliferation. Nevertheless, these anticancer drugs also harm or destroy healthy, normal cells. The patient will consequently have serious adverse effects, and the effectiveness of the therapy will be limited or diminished[20].
The development of a normal, healthy cell into a tumour cell is known as carcinogenesis which involves involving numerous genes and genetic changes. Carcinogenesis is a multiplex process encompassing origination, promotion, and progression[21]. The first step involves the beginning of a permanently altered cell and is frequently linked to a mutation and several start pathways. The initial altered cells grow and manifest as a visible mass of cells during the second stage, which is most likely a benign lesion. Epigenetic elements that influence the proliferation of the started cells are undoubtedly present during the promotion stage. It is not well understood exactly how the second stage of carcinogenesis works.
Mostly benign or non-cancerous cells, or occasionally pre-cancerous cells, are the end result of promotion. When these benign cells transition into neoplastic cells, they experience a few additional genetic changes. The third and final stages of carcinogenesis, which involve the development of malignant tumours from benign non-cancerous tumours, are distinct from the first two steps[22] .
Stem cells play a crucial role in the beginning of carcinogenesis owing to variety of physical, chemical, or biological stimuli, including viruses. The sequential steps are crucial in the malignant transformation of preneoplastic cells because such initiated cells would then be exposed to a promotional factor to accelerate the full neoplastic cell creation[23]. A multicellular animal's carcinogenesis process results from numerous cellular chemical, physical, biological, or genetic alterations. Although mutation is the primary cause of carcinogenesis, a number of additional variables also contribute to its growth. The use of either conventional anticancer drugs or plant derived compounds are available option to reverse or capture the process of carcinogenesis which influence the carcinogenesis process at each stage of cancer development.
- Chapter 3 needs to be rewritten in relation to "phytochemicals". Can phytochemicals solve the drawbacks described in Chapter 4?
We have discussed the progression of conventional anticancer drugs in chapter 3 followed by hindrances in cancer treatment by these conventional anticancer drugs in chapter 4. In chapter 5, we have written about the treatment of cancer using phytochemicals as follows:
Because of the limitations of conventional chemotherapeutic drugs, there is an urgent need for novel cancer treatment. Recently plant-derived agents namely phytochemicals have drawn the attention of researchers as a new treatment modality for cancer owing to various attributes of less adverse effects, action through multiple pathways, and cost-effectiveness [2],[10]. From ancient times, humans have made great use of plants for the treatment of various kinds of ailments including cancer[41]. Since mostly available conventional chemotherapeutic drugs like vincristine, vinblastine, and paclitaxel are plant-derived, the attention of researchers turned towards phytochemicals for the treatment of cancer[42].
According to studies, there are over 250000 plant species in the plant kingdom, but only about 10% of those have been proved their potential as a treatment option for various kinds of diseases depicting that a vast portion of plant species is yet to be explored which can cause a revolution in the treatment of cancer[9]. Various phytochemicals and their derivatives are present in diverse parts of the plants like seeds, flowers, bark, fruit, leaf, embryo, and rhizomes[43]-[44]. In addition, these phytochemicals and their derivatives possess various pharmacological properties like anti-inflammatory, antimicrobial, antifungal, antihypertension, antiaging, antioxidant, immunomodulator, antimalarial, anticancer etc.
Various plant-derived products like flavonoids, alkaloids, terpenes, vitamins, glycosides, minerals, oils, gums, biomolecules and other metabolites (primary or secondary) have proven their anticancer potential owing to various mechanisms; inhibition of cancer cell activating proteins, enzymes like cyclooxygenase, topoisomerase, CDK2, Cdc2, CDK4 kinase, MMP, MAPK/ERK, signalling pathways, activation of DNA repair mechanism, induction of antioxidant action or stimulation of the formation of protective enzymes (Caspase-3, 7, 8, 9, 10, 12)[45]-[46] (Figure 3).
Various preclinical studies have demonstrated that the combination of phytochemicals with conventional anticancer drugs is more efficacious than phytochemicals individually to treat cancer due to the fact that plant derived compounds have low anticancer efficacy than conventional anticancer drugs.
Various phytochemicals having the potential to improve anticancer activity in codelivery include curcumin, resveratrol, genistein, epigallocatechin gallate, allicin, quercetin, thymoquinone, piperine, naringenin, naringin, emodin, luteolin, β-carotene, anthocyanins, berberine, ursolic acid, withaferin A, sulforaphane, colchicine[47],[13] (Figure 4).
Although phytochemicals have huge potential as anticancer drugs these also suffer from various limitations such as low solubility, poor bioavailability, high dose, narrow therapeutic index, fast absorption by the normal cells, high apparent volume of distribution leading to accumulation of drugs in normal cells, high clearance rate, short elimination half life[13],[48]. However, phytochemicals also have the potential to improve the anticancer property of other chemotherapeutic agents by decreasing their adverse effects[49]-[50]. Hence these days, plant-derived drugs or phytochemicals are used in combination with conventional chemotherapeutic agents for efficacious treatment of cancer with low adverse effects[51] .
- Figure 1 also needs to be modified (related to "phytochemicals" and "nanocarrier?"). Information on the occurrence of simple cancers is not required.
Ans: In figure no.1, our aim was to provide the basics of cancer along with its progression. We have now incorporated your valuable suggestions also in Figure no.1. However, we have provided the information about the phytochemicals and nanocarrier in figure number 6 which states that Various nanocarriers like solid lipid nanoparticles, nanostructured lipid carriers, nanoemulsions, polymeric nanoparticles, polymeric micelles, liposomes, dendrimers, carbon nanotubes, metallic nanoparticles, nanoemulsions have been utilized for codelivery of anticancer drugs owing to their ability to entrap the drugs followed by release on targeted site[74]-[75] (Figure 6). Moreover, they also protect the drug molecules from hazardous environmental factors which can cause gastrointestinal degradation of the drugs[76]. The modification in shape, size, and surface properties of nanocarriers can be performed to elicit maximum efficiency which leads to improved drug efficiency, decreased adverse effects, avoidance of multiple drug resistance, and maximization of drugs in targeted cells[77].
Figure 1: Various stages for the progression of cancer along with their treatment approaches
Figure 6: Mechanism of various nanocarriers to target the drug at the cancer site
- Chapter 5 is good, but it is needed to break up the paragraphs and clarify the content for the potential readers.
Ans: Thank you for your valuable suggestion. We have break up the Chapter 5 in paragraphs to clarify the content for potential readers.
- Check the grammar of the words in Figure 2 again.
As per your suggestions, we have made some changes in figure 2.
Figure 2: Various hindrances associated with conventional anticancer drugs
- Overall, I think it will help to understand if I divide the paragraphs and describe them.
Ans: By keeping in mind your valuable suggestion we have break up the manuscript in paragraphs for the interest of the potential readers.
- It would be good to reflect the classification (nanocarriers) in Chapter 6 in Table 2 as well. The contents of Table 2 are good, but the classification is awkward. For example, What does simple nanoparticle (Table 2) mean (not SLN?, NLC?) ?
Ans: Thank you for valuable suggestions of the reviewer. We have classified all the nanocarriers with specific category the table 2 in chapter 6 (Example: simple nanoparticles have been classified as polymeric nanoparticles).

Round 2
Reviewer 1 Report
Only one additional concern. Could you discuss the progress of clinical trials of listed phytochemicals as monotherapy if no co-delivery system? For example, for curcumin, extracellular vesicle has been used to deliver curcumin (NCT01294072).
Author Response
Reviewer 1
- Could you discuss the progress of clinical trials of listed phytochemicals as monotherapy if no co-delivery system? For example, for curcumin, extracellular vesicle has been used to deliver curcumin (NCT01294072)
Ans: Till date, no combination of conventional anticancer drug with phytochemicals is available as anticancer treatment but curcumin in monotherapy is in clinical trials in market as anticancer drugs. In NCT01294072, a phase I randomized clinical trial was conducted to study the ability of plant exosomes to deliver curcumin to normal or colon cancer tissues enrolling 35 participants. The recruitemnt status was recruiting. The primary outcome measure of study were to compare the concentration of curcumin in normal tissues and cancerous tissues after 7 days of ingestion and secondary outcome measures were safety and tolerability of curcumin alone as determined by adverse events after 7 days of enrollement[53].

Reviewer 2 Report
The answers to the previous questions are nice, but the authors made few changes to the text and abstact.
It is recommended that it be published after further revision.
1.Check all abbreviated words in the text. WHO is unexplained and EPR is explained three times.
2.It's not good to start with "therefore" or "however" in the paragraph. Replace them all if possible.
3.Make sure all paragraphs are semantically complete. Overall, the content is too listed.
4. "In a study" appears too many times in the text. It is better to change it to something else in a variety of different expressions.
5. There is an error in the font size in the text.
Author Response
Reviewer 2
1.Check all abbreviated words in the text. WHO is unexplained and EPR is explained three times.
Ans: Thank you for pointing out this. As per your suggestions, we have corrected abbreviations throughout the manuscript.
2.It's not good to start with "therefore" or "however" in the paragraph. Replace them all if possible.
Ans: Thank you for providing insightful and valuable suggestion. As per your suggestion, we have deleted or replaced the above mentioned words throughout the manuscript.
3.Make sure all paragraphs are semantically complete. Overall, the content is too listed.
Ans: As per your suggestion, we have gone through complete manuscript and we made some changes where the paragraphs were not semantically complete.
- "In a study" appears too many times in the text. It is better to change it to something else in a variety of different expressions.
Ans: Thank you for valuable suggestion of the reviewer. As per your suggestions, we have made changes throughout the manuscript.
- There is an error in the font size in the text.
Ans: Thank you for pointing out this point. We have done same font size throughout the manuscript.
